# Euchromatic Histone Lysine Methyltransferase 2 Inhibition Enhances Carfilzomib Sensitivity and Overcomes Drug Resistance in Multiple Myeloma Cell Lines

**DOI:** 10.3390/cancers15082199

**Published:** 2023-04-07

**Authors:** Elisabetta Mereu, Damiano Abbo, Tina Paradzik, Michela Cumerlato, Cecilia Bandini, Maria Labrador, Monica Maccagno, Domenica Ronchetti, Veronica Manicardi, Antonino Neri, Roberto Piva

**Affiliations:** 1Department of Molecular Biotechnology and Health Sciences, University of Turin, 10126 Turin, Italy; 2Department of Physical Chemistry, Rudjer Boskovic Insitute, 10000 Zagreb, Croatia; 3Department of Oncology and Hemato-Oncology, University of Milan, 20122 Milan, Italy; 4Laboratory of Translational Research, Azienda USL-IRCCS di Reggio Emilia, 42123 Reggio Emilia, Italy; 5Scientific Directorate, Azienda USL-IRCCS di Reggio Emilia, 42123 Reggio Emilia, Italy; 6Medical Genetics Unit, Città della Salute e della Scienza University Hospital, 10126 Turin, Italy

**Keywords:** multiple myeloma, proteasome inhibitors, functional screening, EHMT2, drug resistance, combinatorial treatment

## Abstract

**Simple Summary:**

In the last decade, proteasome inhibitors have become a standard treatment for multiple myeloma. However, resistance to combined therapies remains a challenge due to the disease’s complex biology, crosstalk with the bone marrow microenvironment, and toxicities. To address this issue, high-throughput functional approaches are identifying new synthetic lethal interactions. In this study, a library of small-molecule inhibitors was used to identify compounds that could enhance the efficacy of proteasome inhibitors in multiple myeloma. The results suggest that EHMT2 inhibition could be a feasible strategy to increase proteasome inhibitor sensitivity and overcome drug resistance in multiple myeloma patients.

**Abstract:**

Proteasome inhibitors (PIs) are extensively used for the therapy of multiple myeloma. However, patients continuously relapse or are intrinsically resistant to this class of drugs. In addition, adverse toxic effects such as peripheral neuropathy and cardiotoxicity could arise. Here, to identify compounds that can increase the efficacy of PIs, we performed a functional screening using a library of small-molecule inhibitors covering key signaling pathways. Among the best synthetic lethal interactions, the euchromatic histone-lysine N-methyltransferase 2 (EHMT2) inhibitor UNC0642 displayed a cooperative effect with carfilzomib (CFZ) in numerous multiple myeloma (MM) cell lines, including drug-resistant models. In MM patients, EHMT2 expression correlated to worse overall and progression-free survival. Moreover, EHMT2 levels were significantly increased in bortezomib-resistant patients. We demonstrated that CFZ/UNC0642 combination exhibited a favorable cytotoxicity profile toward peripheral blood mononuclear cells and bone-marrow-derived stromal cells. To exclude off-target effects, we proved that UNC0642 treatment reduces EHMT2-related molecular markers and that an alternative EHMT2 inhibitor recapitulated the synergistic activity with CFZ. Finally, we showed that the combinatorial treatment significantly perturbs autophagy and the DNA damage repair pathways, suggesting a multi-layered mechanism of action. Overall, the present study demonstrates that EHMT2 inhibition could provide a valuable strategy to enhance PI sensitivity and overcome drug resistance in MM patients.

## 1. Introduction

Proteasome inhibitors (PIs) have improved the survival of multiple myeloma (MM) patients, and nowadays they represent the backbone of MM therapy in all treatment phases [1]. Proteasome inhibition disrupts the balance between protein production and disposal of misfolded or non-functional proteins, leading to proteotoxic stress, excess activation of the unfolded protein response, and, ultimately, apoptosis. Since MM cells produce vast amounts of secretory proteins, they heavily rely on proper proteasome function [1,2]. To date, three PIs are routinely used in clinics. The first-in-class PI was bortezomib (BTZ), a slowly reversible inhibitor of the β5 catalytic proteasomal subunit. Subsequently, the irreversible inhibitor of β5 site carfilzomib (CFZ) and the first orally administered PI ixazomib (IXZ) were approved [3]. CFZ was approved by the U.S. FDA in 2012 for patients who have received at least two prior lines of therapy and are refractory to other recent drugs [4]. The main advantage of CFZ treatment is its long duration of proteasome inhibition, which has shown effectiveness against relapsed and/or refractory MM patients. Clinical trials have been conducted using CFZ as a single agent or in combination with other compounds [5]. Although some collateral effects have been reported, patients have reported improved health-related quality of life, making CFZ an established new standard of care for patients with relapsed multiple myeloma. Despite the advances driven by proteasome inhibitors, relapses and disease progressions are common among MM patients, suggesting a prominent role in either innate or acquired drug resistance. Moreover, although the toxicity of PI is quite well controlled in clinical settings, they display distinct adverse profiles, imposing limits to their doses. The pleiotropic consequences of proteasome inhibition often result in synergistic or additive activity with other therapeutic protocols, leading toward the identification of novel combination strategies to overcome resistance and broaden the applicability of this class of drugs [6]. Approved combinations for clinical use include dexamethasone, immunomodulatory drugs, SLAMF7/CD38-targeting antibodies, or histone deacetylase (HDAC) inhibitors. Combinatorial treatments have significant advantages over single drugs, such as toxicity reduction, by minimizing the doses and therapeutic effect escalation through synergistic interactions. In the context of drug resistance, attacking multiple targets may reduce or delay recurrences [7].

To identify compounds enhancing the efficacy of PI in the combinatorial setting, we performed a functional screening in the PI-Resistant cell line U266^PIR^ using a library of small-molecule inhibitors covering key targets implicated in a wide variety of signaling pathways. The best synthetic lethal interactions included drugs targeting pathways already known to synergize with PIs and several new compounds. Among these, the euchromatic histone-lysine N-methyltransferase 2 (EHMT2) inhibitor UNC0642 displayed a cooperative effect with CFZ in a large panel of MM cell lines tested (including 4 PIR models). UNC0642 is a substrate-competitive inhibitor of EHMT1/2, which blocks the SET domain from receiving methyl groups from its cofactor, S-adenosyl-methionine (SAM). This results in a potent inhibition of EHMT1/2 activity in vitro and has been shown to reduce H3K9 dimethylation levels in various cellular models [8].

EHMT2 (also known as G9a) is a nuclear histone lysine methyltransferase (HMT) belonging to the Su(var)3-9 family that mainly catalyzes histone H3 lysine 9 (H3K9) mono- and di-methylation, a reversible modification generally associated with transcriptional gene silencing [9]. Structurally, it is composed of a catalytic SET domain, an ankyrin repeats domain, and a nuclear localization signal at the N-terminal region. The SET domain is responsible for the addition of methyl groups on H3, whereas the ankyrin repeats are involved in mono- and dimethyl lysine binding, functioning as a scaffold for the recruitment of other target molecules on the chromatin [9]. In the context of cancer, EHMT2 methyltransferase activity has been linked to repression of tumor suppressors such as TP53 and RUNX3 and reciprocal modulation of EMT-linked genes such as E-cadherin and vimentin [10,11]. Indeed, in addition to its well-established role in gene suppression, EHMT2 is known to act as a co-activator in association with other factors such as the coactivator-associated arginine methyltransferase 1 (CARM1), histone acetyltransferases p300, RNA polymerases, or the mediator complex to activate the expression of genes as p21, β-globin, and to regulate cell migration [9,10]. Finally, EHMT2 is capable of post-translational protein regulation by methylating non-histone proteins such as p53, FOXO1, ATG-12 and even itself [12,13,14,15]. EHMT2 self-methylation was not linked to changes in its catalytic activity but could affect protein interactions and, in turn, modulate the switching between its repressor or activator functions [10,12,16] (Appendix A).

In the present study, we found that EHMT2 expression was increased in bortezomib-resistant MM patients and was correlated to worse survival. CFZ/UNC0642 combinatorial treatment exhibited a favorable cytotoxicity profile toward peripheral blood mononuclear cells and bone-marrow-derived stromal cells. To exclude off-target effects, we proved that UNC0642 treatment reduces EHMT2-related molecular targets and that an alternative EHMT2 inhibitor recapitulated the synergistic activity with CFZ. Finally, we showed that the CFZ/UNC0642 combination significantly perturbs autophagy and the DNA damage repair pathways, suggesting a multi-layered mechanism of action.

## 2. Materials and Methods

### 2.1. Cell Culture Conditions and Reagents

Human multiple myeloma (MM) cell lines KMM-1, U266, OPM-2, KMS28-BM, U266^PIR^, and KMM1^PIR^ were obtained from the Japanese Collection of Research Bioresources Cell Bank (JCRB), National Institutes of Biomedical Innovation, Health and Nutrition, Ibakari, Osaka, Japan; the Leibniz Institute-German Collection of Microorganisms and Cell Cultures GmbH (DSMZ), Braunschweig, Germany; or generated in our lab and authenticated by DNA fingerprinting using the GenePrint system (Promega, Madison, WI, USA). AMO-1^PIR^, RPMI^PIRCFZ^, RPMI^PIRBTZ^, RPMI-8226, and AMO-1 were obtained from the Experimental Oncology Laboratory of St. Gallen, Switzerland [17]. HS-5 (GFP+) cells and RPMI-8226 (tRFP+)-positive cells were derived by transduction with pLKO-GFP and pLemir control vectors, respectively. All cell lines were maintained in RPMI 1640 medium (Euroclone, Pero, Italy), supplemented with penicillin 100 U/mL, streptomycin 100 μg/mL, and L-glutamine 2 mM (Gibco) 10% fetal bovine serum (FBS, Euroclone) and grown at 37 °C in humidified atmosphere with 5% CO_2_. Peripheral blood mononuclear cells (PBMCs) from healthy blood donors were provided by the local Blood Bank (Fondazione Strumia) and isolated on a Ficoll–Hystopaque density gradient. For the co-cultures experiment, KMS28-BM cells were seeded on HS-5 bone marrow stromal cell line expressing GFP protein. The percentage of live multiple myeloma cells was measured over time using hCD138-APC antibody (clone: 44F9; 130-117-395, Miltenyi Biotec). Carfilzomib (PR-171, Catalog No. S2853), UNC0642 (HY-13980, Catalog No. S7230), and A366 (Catalog No. S7572) were obtained from Selleckchem and bortezomib (PS-341) from Millennium Pharmaceuticals.

### 2.2. Cell-Growth Luminescence Assay

The cell-growth assay was performed using CellTiter-Glo^®^ Luminescent Cell Viability Assay (Catalog No. G7570, Promega) to measure cell-growth rate, according to manufacturer instructions. Briefly, 10 μL of cells were mixed with an equal volume of CellTiter-Glo^®^ Reagent solution in a 384-well white plate, in technical duplicates. The plate was shaken for 2 min and then incubated in the dark at room temperature for 10 min. Luminescence was measured using BioTek SynergyTM 2 Multi-Mode Microplate Reader (Bio-Tek Instruments, Winooski, VT, USA).

### 2.3. Drug Screening

Primary screening was performed using the target-selective inhibitor library (Catalog No. L3500-Selleck Chemicals). The library was assembled with 320 small-molecule inhibitors covering 123 key targets implicated in a wide variety of signaling pathways. The compound library in a 96-well plate format was diluted in DMSO or water at a working solution of 10 mM according to manufacturer instructions. U266^PIR^ cell treatment was performed in duplicate using 4 compound concentrations (10 μM, 1 μM, 100 nM, and 10 nM). After two hours, cells were treated with a sublethal dose of carfilzomib (20 nM) or with control DMSO. Cell viability was assessed using Cell TiterGlo (Promega) luminescence assay performed at day 0 and 72 h post-treatment, in duplicate. Growth rate (GR) was calculated as the ratio between luminescence at day 3 and luminescence at day 0, normalized to DMSO-treated cells. The combined drug effect was determined by excess over Bliss (EOB) analysis of the GR value for all concentrations, according to the following formula: EOB = (1 − GR(combination)) − (1 − GR(CFZ)) − (1 − GR(drug)) + (1 − GR(CFZ))(1 − GR(drug)) [18]. The most synergistic drugs (TOP 40) with EOB ≥ 0.4 were chosen for subsequent analysis. The secondary screening was performed in 4 independent MM cell lines (U266, KMM-1, AMO-1, KMS28-BM) using the same conditions of the primary screening. A synergy score was assigned to each compound according to the EOB value obtained in all cell lines. The most synergistic candidate (TOP10) showed the highest score in more than 50% of MM cell lines tested in secondary screening.

### 2.4. Analysis of Apoptosis

Apoptosis was measured by flow cytometry after staining with tetrametylrodamine methyl ester (TMRM; Molecular Probes, Eugene, OR, USA) or Annexin V-FITC Kit (Catalog No. 130-092-052, Miltenyi Biotec, Bergisch Gladbach, Germany), according to the manufacturer’s instructions. CD138+ cells were identified by anti hCD138-APC antibodies (clone: 44F9; 130-098-197, Miltenyi Biotec). Data were acquired using a FACSCelesta cytofluorimeter and processed with BDIVA 8.0 software (BD Biosciences, Franklin Lakes, NJ, USA).

### 2.5. Proteasome-Inhibitor-Resistant (PIR) Cell Reconditioning Protocol

To induce AMO-1^PIR^, RPMI^PIRCFZ^, and RPMI^PIRBTZ^ to express full resistance and eliminate sensitive clones, after thawing, cells were seeded 1 × 10^5^/mL and treated with two rounds of the corresponding PI. For RPMI^PIRCFZ^, 50 nM of CFZ was used both for the first and second reconditioning step. For RPMI^PIRBTZ^ and AMO-1^PIR^, 50 nM of BTZ was used for the first reconditioning step and 70 nM for the second one. Sensitive cells were treated in parallel with a lethal dose of CFZ and BTZ as control. Cell viability was monitored with flow cytometry–TMRM staining.

### 2.6. 3D Co-Culture

The 3D scaffolds were populated according to protocol described by Belloni et al. [19] and adapted to MM cells growth in static conditions. Briefly, scaffold discs were cut from SpongostanTM sheets (Cod. MS0005, Ethicon, Inc., Somerville, NJ, USA) using a sterile 4 mm biopsy punch and then pre-seeded with stromal cell line HS-5-GFP-positive (200,000 cells/scaffold). Twenty-four hours later, RPMI-8226-tRFP-positive cells (500,000 cells/scaffold) were added to the scaffold and cultivated in a 24-well plate in 1 mL RPMI culture medium. Every 24 h up to 3 days, scaffolds were transferred into a new well to remove detached cells. Drug treatments were performed 72 h post-RPMI-8226-tRFP seeding using sublethal doses of CFZ (6 nM), UNC0642 (10 μM), or the combination of the two drugs. Six days after treatment, the ratio between stromal and tumoral cells was analyzed using FACS analysis following scaffold lysis with LiberaseTM (Cod. 05401020001, Merck Millipore, Burlington, MA, USA) (25 μg/mL).

### 2.7. Purification of Total RNA and Reverse Transcription–Quantitative Polymerase Chain Reaction (RT-qPCR)

Total RNA was extracted using the RNeasy Mini Kit (Qiagen, Hilden, Germany) according to the manufacturer’s instructions. cDNA was obtained from total RNA, previously treated with RQ1 RNase-free DNase (Catalog. No. M6101, Promega), using OneScript^®^ Plus cDNA Synthesis Kit (Cat. No. G236, Applied Biological Materials Inc, Richmond, VA, Canada) following the manufacturer’s instructions. Quantitative PCR reactions were performed in 384-well plates with a C1000TM Thermal Cycler (Bio-Rad Laboratories, Hercules, CA, USA) using the BlastTaq Green 2× qPCR Master mix (Cat. No. G891, Applied Biological Materials Inc., Richmond, VA, Canada) according to manufacturer’s instructions. The PCR cycling conditions were the following: 95 °C for 3 min, followed by 40 cycles at 95 °C for 15 s and 60 °C for 1 min. The oligonucleotide primer pairs used for RT-qPCR were designed with PrimerBLAST (http://www.ncbi.nlm.nih.gov/tools/primer-blast/ accessed on 26 February 2023) and synthesized upon request by Eurofins, Luxembourg. To confirm amplification specificity, the PCR products were subjected to the analysis of the melting curve. All PCR assays were performed in triplicate, and the average Ct (cycles to threshold) was used for the comparative Ct method [20]. Quantification of GAPDH levels served as an endogenous control.

The following table lists the primer sequences used in the present study for RT-qPCR (Table 1): 

### 2.8. Western Blotting

Protein extracts were obtained using lysis buffer composed of 20 mM Tris-HCl (pH 7.4), 150 mM NaCl, 5 mM EDTA, 1% Triton X-100, 1 mM PMSF, 10 mM NaF, 1 mM Na_3_VO_4_, and protease inhibitor cocktail (Cod. 11836153001, Roche, Basilea, Switzerland). Total protein concentrations were measured using Bio-Rad DC protein assay kit (Cod. 5000111, Bio-Rad). Equal amounts of protein lysates were resolved by SDS-PAGE (Mini-PROTEAN TGX Stain-free Precast Mini Gels, Bio-Rad) and transferred to the nitrocellulose membrane. The membranes were first blocked for 1 h at room temperature with 5% low-fat milk in phosphate-buffered saline (PBS) solution with 0.1% Tween 20 and then incubated overnight with the primary antibodies diluted in BSA 5% + NaN_3_ at room temperature. After 3 washes, membranes were incubated with the secondary antibody, diluted in PBT 5% low-fat milk for 1 h at room temperature. After 3 washes, the immune complexes were detected using Immobilon Western Chemiluminescent HRP Substrate (Cod. WBKLS0500, Merk). Band quantification was performed using Image Lab software (Bio-Rad), calculating the ratio between target protein expression intensity with its housekeeping gene and normalizing all values to control conditions.

The following primary antibodies were used in this study (Table 2):

### 2.9. Gene Expression Data

The expression and prognostic value of EHMT2 messenger RNA (mRNA) in terms of overall survival and progression-free survival were determined using publicly available RNA-sequencing (RNAseq) data from newly diagnosed MM patients from the Multiple Myeloma Research Foundation’s (MMRF) CoMMpass study (https://research.themmrf.org/ accessed on 26 February 2023; release IA18). Data were analyzed with GraphPad prism 9.1 to retrieve mean EHMT2 expression (Transcript per million, TPM) and standard deviation.

### 2.10. Statistical Analysis

Statistical analyses and IC50s were performed with GraphPad Prism 9.1 (GraphPad Software Inc., San Diego, CA, USA). Statistical significance of differences observed was determined by Student’s *t*-test and one-way/two-way ANOVA; differences were considered significant when the *p*-value was <0.05 (*), <0.01 (**), <0.001 (***), or <0.0001 (****).

## 3. Results

### 3.1. Drug-Library Screening in Multiple Myeloma Cell Lines Identifies EHMT2 Inhibition as Synthetic Lethal to the Proteasome Inhibitor Carfilzomib

To identify compounds that can increase the efficacy of CFZ in the combinatorial setting, we performed a functional screening using covering 123 key targets implicated in a wide variety of signaling pathways. The primary screening was performed in the U266^PIR^ cell line, previously generated in our laboratory to be cross-resistant to BTZ and CFZ [21]. Cells were exposed to drugs at four concentrations (10 μM, 1 μM, 100 nM, and 10 nM) in the presence or absence of a sublethal concentration of CFZ and analyzed after 72 h to calculate growth rate (GR) and excess over Bliss (EOB) score (Figure 1). An arbitrary EOB cut-off (0.4) was used to define the top 40 synergistic compounds that were further validated in four PI-sensitive MM cell lines (AMO-1, KMM-1, KMS28-BM, U266). The secondary screening identified 10 prominent synthetic lethal targets. Among these, we found drugs targeting pathways already known to synergize with PIs, such as PI3K/AKT/mTOR, BTK, and β-catenin [22,23,24]. The histone methyltransferases EHMT2 inhibitor UNC0642 displayed synergistic interaction in five out of five MM cell lines; thus, it was selected for further validation. 

### 3.2. EHMT2 Expression Is Increased in Bortezomib-Resistant Multiple Myeloma Patients and Correlates to Worse Survival

To investigate the significance of EHMT2 protein in multiple myeloma pathogenesis, we computationally evaluated its mRNA and protein expression. First, the DepMap database (https://depmap.org/portal/ accessed on 26 February 2023) revealed that mRNA and protein levels of EHMT2 in MM cell lines were comparable to other type of tumor cell lines. Next, we considered EHMT2 expression in two microarray databases including a cohort of 20 MGUS, 33 smoldering myeloma, 170 MM, 36 plasma cell leukemia (PCL) patients, and 9 healthy donors [25,26]. The analysis did not highlight any significant correlation of EHMT2 expression with disease progression. Finally, to better elucidate the clinical and biological relevance of EHMT2 expression, we took advantage of a large cohort of MM patients enrolled in the Multiple Myeloma Research Foundation CoMMpass study (https://research.themmrf.org/ accessed on 26 February 2023). Based on RNA sequencing data, we stratified patient populations in EHMT2-high and EHMT2-low expression, and we found that high EHMT2 expression correlates with worse overall survival (*p* = 0.0001) and progression-free survival (*p* = 0.0109) (Figure 2a,b). Interestingly, CoMMpass dataset analysis revealed that EHMT2 expression was significantly higher (*p* = 0.0004) in patients who experienced disease progression/relapse after bortezomib treatment compared to patients who completed the regimen (Figure 2c), thus suggesting that EHMT2 could be implicated in the processes of drug resistance. We also assessed EHMT1 and EHMT2 mRNA and protein levels in MM cell lines. However, we did not detect any significant difference in expression between PI-resistant and sensitive MM cell lines (Appendix A).

### 3.3. EHMT2 Inhibition Synergically Increases Proteasome Inhibitor-Mediated Cell Death in PI-Resistant and PI-Sensitive Multiple Myeloma Cell Lines

To test the significance of the screening results, five PI-resistant (U266^PIR^, KMM1^PIR^, AMO1^PIR^, RPMI^PIRBTZ^, RPMI^PIRCFZ^) and six PI-sensitive (U266, KMM-1, KMS28-BM, AMO-1, RPMI-8226, OPM-2) MM cell lines were treated with sublethal doses of CFZ (Appendix A), UNC0642 (Appendix A) or with a combination of the two drugs (Figure 3a and Appendix A). We observed that UNC0642/CFZ combinatorial treatment significantly increased cell death compared to single agents in all cell lines tested. To define the extent of the synergy in PI-resistant cell lines, we performed a UNC0642/CFZ dose–response matrix in AMO-1^PIR^, RPMI^PIRBTZ^, and RPMI^PIRCFZ^, detecting cell-growth inhibition by luminescence growth assay at 72 h post-treatment. Matrix analysis with the SynergyFinder web application (https://synergyfinder.org/ accessed on 26 February 2023) could define AMO-1^PIR^ as synergistic and RPMI^PIR^ as additive responders to UNC0642/CFZ combination (Figure 3b–d).

### 3.4. Combining EHMT2 and Proteasome Inhibition Is Not Toxic to Normal Cells and It Is Effective in the Presence of Bone Marrow Milieu

To exclude general toxicities of UNC0642/CFZ combination in human normal cells, peripheral blood mononuclear cells (PBMCs) purified from four healthy donors were treated with increasing concentrations of UNC0642 in the presence of a sublethal dose of CFZ. FACS analysis of Annexin V-PI staining at 72 h post-treatment demonstrated a favorable cytotoxicity profile of the combinatorial treatment in PBMCs, as compared to the MM cell line KMS28-BM (Figure 4a). Furthermore, UNC0642/CFZ combination exhibited low cytotoxicity towards the bone-marrow-derived stromal cell line HS-5 (Figure 4b). To exclude that the bone marrow milieu could impair combinatorial treatment efficacy with secretion of pro-survival factors, we co-cultured KMS28-BM myeloma cells on a layer of GFP-positive HS-5 cells. We demonstrated that UNC0642/CFZ combination progressively decreased the percentage of KMS28-BM cells, thus also exhibiting a selective cytotoxicity in the presence of stromal cells (Figure 4c). Moreover, to fully investigate MM cells response to the drugs, we exploited 3D culturing scaffolds that efficiently reproduce multiple myeloma–bone marrow interactions [27]. Even though the 3D microenvironment reduced sensitivity to CFZ of RPMI-8226-tRFP-positive cells co-cultured with HS-5-GFP-positive cells, the UNC0642/CFZ combination resulted highly selective in killing MM cells in the 3D microenvironment (Figure 4d). These data confirmed the feasibility and safety of the suggested combinatorial treatment.

### 3.5. Pathway Specificity: Investigation of Molecular Markers Associated with EHMT2 Inhibition

To investigate the specificity of EHMT2 inhibition in combination with CFZ, we first verified the regulation of canonical EHMT2 targets such as MYC that has been found to be modulated in multiple type of cancers [28,29]. AMO-1^PIR^, RPMI^PIRBTZ^, and HS-5 cells were treated with increasing concentrations of UNC0642 and tested by RT-qPCR and Western blotting. We observed decreased MYC mRNA and protein levels in the synergistic AMO-1^PIR^ cell line. On the contrary, weak and non-responder cell lines (RPMI^PIRBTZ^ and HS-5) failed to display a significant perturbation of MYC expression (Figure 5a–c). Next, we analyzed the effects of EHMT2 inhibition on H3K9 mono and di-methylation [30]. We confirmed that UNC0642 reduced both mono- and di-methylation levels in all three cell lines tested (Figure 5d). Remarkably, we observed that UNC0642 treatment led to a striking increase in EHMT2 protein levels, while it did not affect EHMT1 or EHMT2 transcripts in all cell lines tested (Figure 5a–d and Appendix A), thus suggesting a post-translational stabilization induced by the drug protein complex. Finally, CFZ was tested in combination with the alternative EHMT2 inhibitor A366. A-366 has low structural similarity to UNC0642; however, it shows an analogous binding mode to EHMT2 and equivalent inhibition of H3K9 dimethylation [31,32]. We observed that A-366 displayed synergistic activity with CFZ in PI-sensitive (Figure 5e) and PI-resistant cells (Appendix A), thus phenocopying UNC0642 action and supporting its specificity in inhibiting the EHMT2/EHMT1 complex.

### 3.6. UNC0642/CFZ Synergistic Interaction Is Not Dependent on Cell-Cycle Perturbation

It is known that EHMT2 affects the cell cycle through several mechanisms such as regulation of the mTOR pathway [33], hypoxia-mediated RUNX3 [34], and IRF-4/MYC axis suppression [35], thus resulting in modified cyclin expression and mitotic marker phosphorylation [36,37]. To elucidate the mechanisms of UNC0642/CFZ synergistic interaction, we monitored the cell cycle in AMO-1^PIR^ and RPMI^PIRBTZ^ cells treated with UNC0642, CFZ, or with the two agents. At 48 h post-treatment, AMO-1^PIR^ showed S-phase reduction and G2/M increase, mainly as a consequence of the CFZ effect. In contrast, RPMI^PIRBTZ^ displayed a significant reduction in S and G2/M phases associated with a G0/G1 arrest, as a consequence of the combinatorial treatment effect (Appendix A). Cell viability analysis at 72 h confirmed the synergistic effect of UNC0642/CFZ in the AMO-1^PIR^ and not in RPMI^PIRBTZ^ (Appendix A). These data suggest that the cytotoxic effect of UNC0642/CFZ combination is not a consequence of cell-cycle perturbation.

### 3.7. Autophagy and DNA Damage Pathways Are Activated by UNC0642/CFZ Combinatorial Treatment

It is well established that EHMT2 inhibits autophagy by several mechanisms, such as the transcriptional repression of autophagy genes, the regulation of mTOR pathway, and the post-translational modification of ATG-12 [14,38,39]. Moreover, it has been recently suggested that EHMT2 inhibition and proteasome inhibition boost autophagic cell death of MM cells through c-MYC downregulation [29]. To investigate the molecular mechanism of UNC0642/CFZ synergistic interaction and avoid cell-death-related confounding effects, we immunoblotted proteins extracted 24 h post-treatment when cells showed equal levels of viability. We observed that UNC0642 strongly induced autophagic markers, as demonstrated by increased LC3-II expression, with CFZ likely exerting a supporting role through ATG-12 accumulation. Interestingly, the strong-responder AMO-1^PIR^ cells were more affected by EHMT2 inhibition than weak-responder RPMI^PIRBTZ^, as demonstrated by increased LC3 accumulation (Figure 6a) and decreased mTOR pathway substrates such as phospho-P70 S6K and phospho-4E-BP1 (Figure 6b). Notably, inhibition of the mTOR pathway was mainly correlated with CFZ treatment, while LC3 accumulation was a consequence of EHMT2 inhibition. However, considering the wide variety of mechanisms in which EHMT2 and the proteasome are implicated, it is unlikely that only one pathway could recapitulate the synergistic phenotype. Remarkably, EHMT2 is involved in DNA damage response together with the replication protein A (RPA) complex and by early recruitment of 53BP1 and BRCA1 to the DNA damage sites [40,41]. Moreover, through its capability to methylate non-histone substrates, it has been shown that EHMT2 controls both the activation and repression of PLK1 and p53 proteins, regulating cell-cycle entrance during the DNA damage conditions [42]. Therefore, we decided to focus our attention on the DNA damage response pathway. Immunoblotting DNA damage markers highlighted a stronger accumulation of p-H2A.X and cleaved PARP following the combinatorial treatment, confirming an activation of the pathway. Phosphorylation of ATM is mainly increased under CFZ treatment conditions, suggesting a prominent role of proteasome inhibition in the activation of the DNA damage response. The increased 53BP1 level following treatments suggested a prevalence of non-homologous end joining (NHEJ) repair over homologous recombination repair (HRR), which could lead to error accumulation and apoptotic cell death (Figure 6c).

## 4. Discussion

Even though PIs have led to substantial outcome improvements in MM patients, the development of novel combination strategies is needed to overcome drug resistance. Through an unbiased drug-library screening, the present study identified the histone methyltransferase EHMT2 as a new synthetic lethal target to PI, effective in both PI-sensitive and -resistant MM cell lines. We showed that the combined targeting of EHMT2 and proteasome triggers synergistic inhibition of human MM in the presence of the bone marrow milieu, with low toxicity to normal human cells. We demonstrated that both the autophagic and the DNA damage pathways are perturbed by the combination, suggesting a pleiotropic effect on multiple cellular processes.

Functional studies using high-throughput screenings are a common method to identify new drug combinations. However, genetic methods such as RNA interference or CRISPR screens could have limitations, such as incomplete target knockdown, off-targets mRNA degradation, induction of DNA damage, or aberrant splicing [43,44,45]. In contrast, drug screenings are ready for validation and translation into the clinic, offering an opportunity for drug repurposing and faster discovery of synergistic drug combinations [45].

To identify targets that synergize with PIs, we performed a drug-library screening, exploiting multiple myeloma cell lines cross-resistant to the proteasome inhibitors bortezomib, carfilzomib, and ixazomib [17,21]. Specifically, U266^PIR^ cells were exposed to drugs in the presence or absence of a sublethal concentration of CFZ. Top candidates were further validated in a secondary screening including several PI-sensitive and -resistant cell lines. The efficacy of the screening is supported by the identification of pathways already known to synergize with PIs, such as PI3K/Akt/mTOR, BTK, and β-catenin [6]. Notably, the EHMT2 inhibitor UNC0642 and other epigenetic drugs ranked in the final list of candidates, highlighting the relevance of epigenetic dysregulation in the context of proteasome inhibition and multiple myeloma pathogenesis (Bandini, C. et al., manuscript in preparation).

Multiple studies have shown that elevated expression of EHMT2 is a common feature in various cancers and hematological malignancies [10]. EHMT2 overexpression has been mainly attributed to transcriptional dysregulation or high copy number and has been linked to advanced tumor stage, metastasis, poor prognosis, and drug resistance. Inhibition of EHMT2 has been shown to modulate tumor suppressor genes, drug metabolism, DNA repair, and cell survival pathways as well as to increase radiosensitivity [46]. Recently, De Smedt et al. reported that EHMT2 inhibition by BIX01294 promotes autophagy-associated apoptosis and boosts proteasome-inhibitor-mediated cell death in multiple myeloma [29], thus independently supporting the validity of our data. We demonstrated that EHMT2 inhibition also increases PI cytotoxicity in PI-resistant multiple myeloma cell lines, and we found that the level of this interaction is cell-line-dependent, ranging from additive to synergistic. Moreover, comparing the efficacy of several PIs highlighted the superior cytotoxicity of the CFZ/UNC0642 combination, thus suggesting that an irreversible and long-lasting proteasome inhibition is required for a synergistic effect. Molecular and biochemical analyses established that strong, weak, and non-responder cell lines had a similar reduction in H3K9 methylation levels upon UNC0642 treatment. In contrast, more prominent MYC downregulation and LC3 accumulation were observed in strong responders compared to weak-responder cell lines. Unexpectedly, UNC0642 treatment induced EHMT2 protein accumulation in all cell lines tested. EHMT2 upregulation was not recapitulated at the mRNA level, thus suggesting a possible stabilization of the EHMT2 protein as a consequence of drug binding and inhibition of its auto-methylation activity. Interestingly, cell-cycle analysis of PI-resistant cell lines revealed that, upon combination treatment, weak-responder cells underwent a significant S- and G2-phase reduction, as strong responders did, that nevertheless was not translated into cell death. Since both strong and weak responders have a similar level of DNA damage markers (increased of cleaved PARP and p-H2A.X), it is possible to speculate that cell-cycle arrest is a putative strategy to avoid DNA damage-related apoptosis and resist the treatment. Accordingly, one of the main features of persister cancer cells is a slow proliferative rate that can arise after treatment [47]. Recently, it was demonstrated that colorectal cancer cells can survive chemotherapy by entering into a diapause-like, slow proliferation state [48]. Further experiments, such as cell-cycle regulator analyses and rescue experiments, are needed to address this issue.

It has been previously described that EHMT1/2-targeting in MM promotes autophagy-mediated cell death and increases the cytotoxicity of proteasome inhibitors [29]. Even though our findings confirmed that EHMT2 inhibition is responsible for triggering autophagy, as demonstrated by LC3 accumulation, we observed that UNC0642 effects on cell viability were rarely significant. Our data reveal that the autophagic process is not sufficient to trigger MM cell death and suggest that the concurrent activation of the DNA damage response is responsible for the synergistic effect of CFZ/UNC0642 combination. Moreover, since we described that this interaction is highly variable in PI-resistant cell lines, preclinical ex vivo studies are encouraged to predict whether the patient will be sensitive to the suggested combination.

## 5. Conclusions

The present study provides new insights into the potential of targeting EHMT2 as a combination strategy with PIs in MM, particularly for patients with drug-resistant disease. Further investigations are needed to confirm these findings and to translate them into clinical practice.

## Figures and Tables

**Figure 1 cancers-15-02199-f001:**
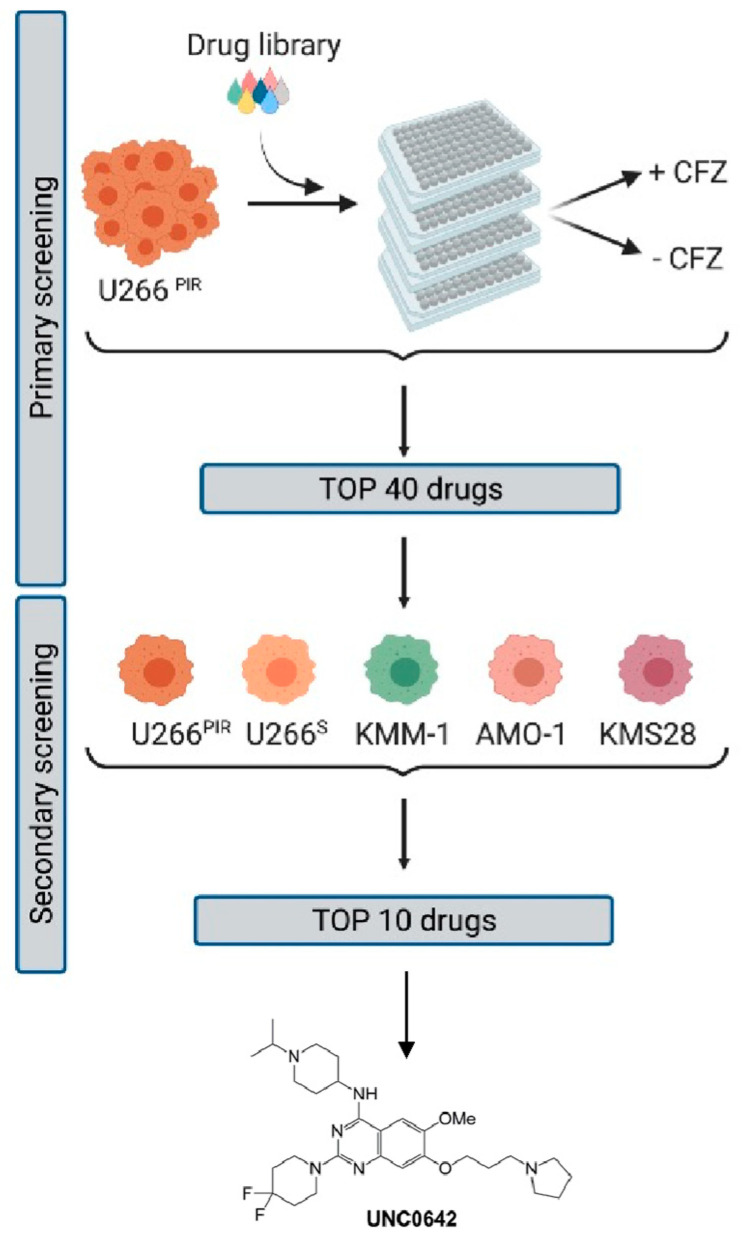
Drug-library screening in multiple myeloma cell lines identifies EHMT2 inhibition as synthetic lethal to the proteasome inhibitor carfilzomib. Experimental design of the drug-library screening. Primary screening was performed on U266^PIR^ cell line. The top 40 candidates were selected using excess over Bliss (EOB) value (cut-off ≥ 0.4) and subjected to a secondary screening that was performed on U266^PIR^, U266, KMM-1, AMO-1, and KMS28-BM cell lines. The EHMT2 inhibitor UNC0642 ranked among the top ten candidates, being synergistic with CFZ in five out of five cell lines at two or more different concentrations (Created by BioRender.com accessed on 26 February 2023).

**Figure 2 cancers-15-02199-f002:**
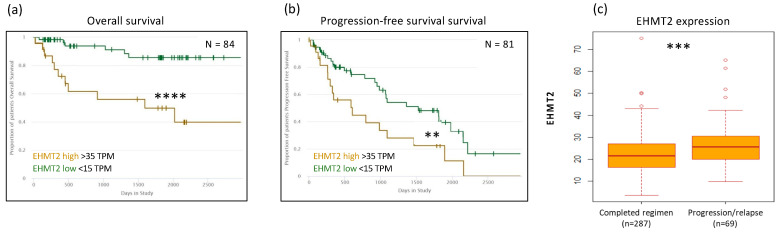
EHMT2 expression is increased in bortezomib-resistant MM patients and correlates to worse survival. (**a**) MMRF CoMMpass dataset IA18 analysis showing significant correlation between EHMT2 expression (RNAseq, TPM) and overall survival (*n* = 84, *p* = 0,0001) and (**b**) progression-free survival (*n* = 81, *p* = 0.0109). One standard deviation from EHMT2 mean expression (TPM) was used to define the sub-populations as low (green line) or high (yellow line) expression. (**c**) Box plots of gene expression level in 287 MM cases that reached a completed regimen in comparison to 69 MM patients who experienced disease progression/relapse after bortezomib treatment (CoMMpass dataset). Differential expression was tested by the Wilcoxon rank-sum test with continuity correction (*p* = 0.0004104). ** *p* < 0.01; *** *p* < 0.001; **** *p* < 0.0001. TPM, transcripts per million.

**Figure 3 cancers-15-02199-f003:**
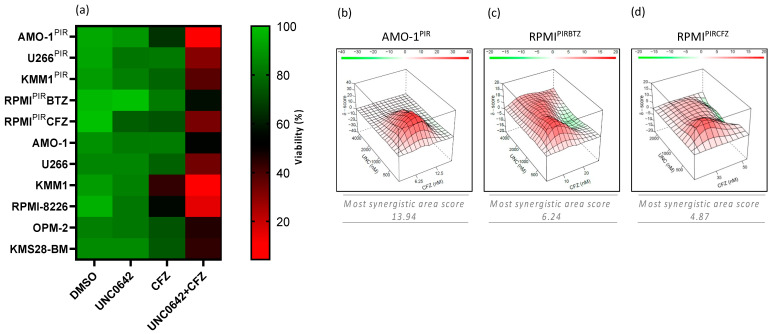
EHMT2 inhibition synergically increases proteasome inhibitor-mediated cell death in PI-resistant and PI-sensitive multiple myeloma cell lines. (**a**) Heatmap representing cell viability after single and combinatorial treatment in U266^PIR^-, KMM1^PIR^-, AMO1^PIR^-, RPMI^PIRBTZ^-, and RPMI^PIRCFZ^-resistant cell lines and U266-, KMM-1-, KMS28-BM-, AMO-1-, RPMI-8226-, and OPM-2-sensitive cell lines. Analysis was performed 72 h post-treatment with FACS or the CellTiterGlo assay (*n* = 3). (**b**–**d**) SynergyFinder analysis of UNC0642/CFZ dose–response matrixes. Percentage of inhibition was retrieved with the CellTiter Glo assay 72 h post-treatment and normalized to the luminescent signal measured at day 0. Synergy score was calculated with Bliss.

**Figure 4 cancers-15-02199-f004:**
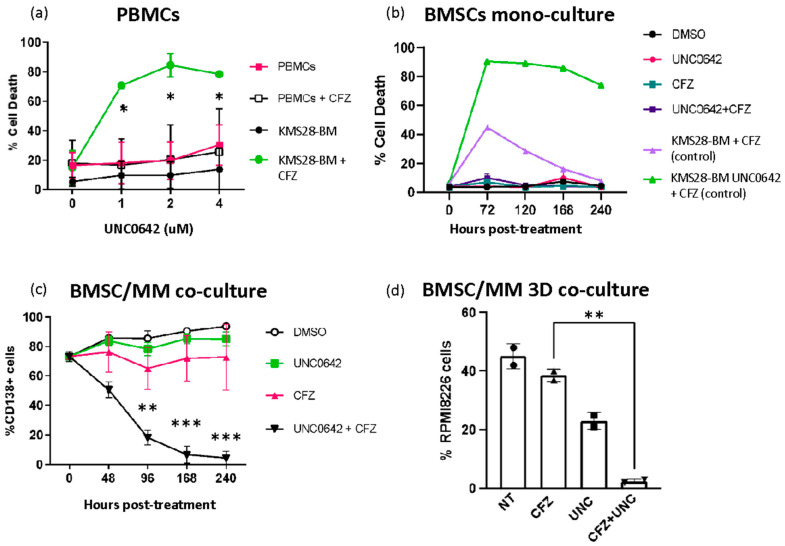
UNC0642/CFZ combination is not toxic for normal cells, and it is effective in the presence of bone marrow milieu. (**a**) PMBCs were extracted from four healthy donors. Combinatorial treatments with increasing doses of UNC0642 together with 2.5 nM CFZ did not affect the viability of PBMCs compared to myeloma KMS28-BM control cells. Annexin V-PI staining was performed 72 h post-treatment. (**b**) GFP+ bone marrow stromal cells (HS-5) were seeded at 50,000/mL and subjected to single or double treatment. KMS28-BM cells were plated separately and used as a control. Cell death was monitored with FACS (PI-staining or TMRM-staining) at the indicated time points. (**c**) The KMS28-BM cell line was cultured on a monolayer of GFP+ HS-5 stromal cells and treated with the indicated doses of UNC0642 and CFZ. Cell viability was estimated by FACS analysis by staining with CD138-APC antibody, and the ratio with GFP+ HS-5 was calculated at the indicated time points. Statistical significance was calculated with the two-way ANOVA test (*n* = 7). (**d**) 3D scaffolds were populated with HS-5 cells (GFP+) and RPMI-I8226 cells (tRFP+) and treated with CFZ, UNC0642, or with the combination. RPMI-8226 cells were specifically detected using FACS analysis 6 days after treatment (*n* = 2 independent replicates) (unpaired t-test CFZ vs. CFZ + UNC0682 *p*-value = 0.0019). * *p* <0.05; ** *p* < 0.01; *** *p* < 0.001.

**Figure 5 cancers-15-02199-f005:**
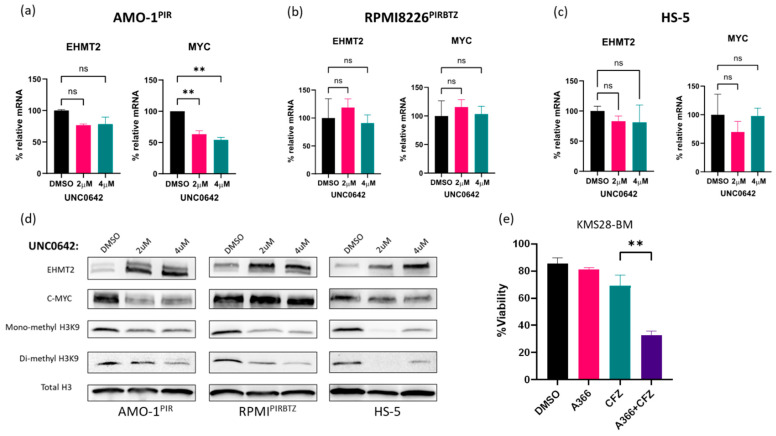
Pathway specificity: investigation of molecular markers associated with EHMT2 inhibition. EHMT2 and MYC mRNA levels in (**a**) AMO-1^PIR^, (**b**) RPMI^PIRBTZ^, and (**c**) HS-5, 24 h post-UNC0642 treatment. Data were normalized on DMSO control (*n* = 2 in AMO-1^PIR^ or *n* = 3 in RPMI^PIRBTZ^ and HS-5). Error bars represent SD. (**d**) Representative immunoblotting of c-MYC, EHMT2, and methylated H3K9 in AMO-1^PIR^, RPMI^PIRBTZ^, and HS-5, 24 h post-UNC0642 treatment. Total H3 was used for protein loading normalization. (**e**) A366 inhibitor (10 uM) recapitulated the synergistic phenotype with CFZ (2.5 nM) in KMS28-BM cell line. Cell viability was monitored with FACS (TMRM staining) 72 h post-treatment. ** *p* < 0.01; ns, not significant.

**Figure 6 cancers-15-02199-f006:**
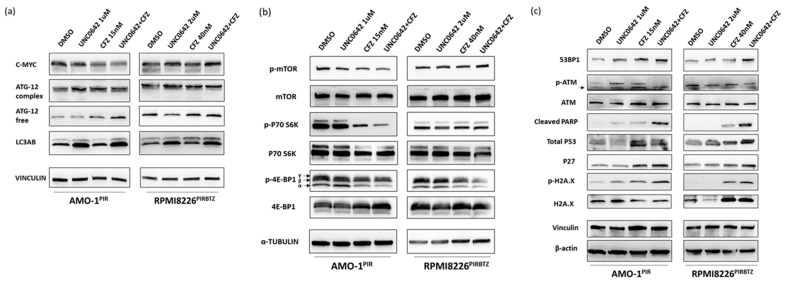
Autophagy and DNA damage pathways are activated by UNC0642/CFZ combinatorial treatment. (**a**) Immunoblotting of autophagic markers, (**b**) mTOR substrates, and (**c**) DNA damage markers in AMO-1^PIR^ and RPMI-8226^PIRBTZ^ treated with UNC0642, CFZ, or the combination at 24 h post-treatment. Vinculin, tubulin, and β-actin were included for protein loading normalization.

**Table 1 cancers-15-02199-t001:** Primer sequences used in the present study.

Gene Name	Sequence
MYC	Fw: GGACCCGCTTCTCTGAAAGG Rw: TAACGTTGAGGGGCATCGTC
EHMT2	Fw: CTGTCAGAGGAGTTAGGTTCTGC Rw: CTTGCTGTCGGAGTCCACG
EHMT1	Fw: CCTCGACTCGGAAAAACCCA Rw: AGTTGGGGTCAATTCCGTCC
GAPDH	Fw: TCTTTTGCGTCGCCAGCCGAG Rw: TGACCAGGCGCCCAATACGAC

**Table 2 cancers-15-02199-t002:** Primary antibodies used in the present study.

Antibody	Species	Source
Vinculin	mouse	SAB4200080, Sigma
α-tubulin	mouse	clone B-5-1-2, T5168, Sigma-Aldrich
β-Actin	mouse	#sc69879, SCB
P27/Kip1	mouse	#610241, BD Biosciences
P53	mouse	#2524S, Cell Signaling Technology
G9a/EHMT2	rabbit	#3306S, Cell Signaling Technology
Histone H3	rabbit	#4499T Cell Signaling Technology
Mono-methyl-histone H3 (Lys9)	rabbit	#14186T Cell Signaling Technology
Di-methyl-histone H3 (Lys9)	rabbit	#4658T Cell Signaling Technology
Cleaved PARP	rabbit	#9532 Cell Signaling Technology
ATG12	rabbit	#4180T Cell Signaling Technology
LC3A/B	rabbit	#12741T Cell Signaling Technology
53BP1	rabbit	#4937S Cell Signaling Technology
p-P70-S6 Kinase (Thr389)	rabbit	#9234T Cell Signaling Technology
P70-S6 Kinase	rabbit	#9202L Cell Signaling Technology
p-mTOR (Ser2448)	rabbit	#5536T Cell Signaling Technology
mTOR	rabbit	#2983T Cell Signaling Technology
p-4E-BP1 (Thr37/46)	rabbit	#2855T Cell Signaling Technology
4E-BP1	rabbit	#9644L Cell Signaling Technology
p-H2A.X (Ser139)	rabbit	#9718T Cell Signaling Technology
H2A.X	rabbit	#2595 Cell Signaling Technology
p-ATM (Ser1981)	rabbit	#5883T Cell Signaling Technology
ATM	rabbit	#2873 Cell Signaling Technology
c-Myc	rabbit	#5605S Cell Signaling Technology

## Data Availability

Not applicable.

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
