# Peer review of "Euchromatic Histone Lysine Methyltransferase 2 Inhibition Enhances Carfilzomib Sensitivity and Overcomes Drug Resistance in Multiple Myeloma Cell Lines"

_cancers, 2023, doi:10.3390/cancers15082199_

Round 1

Reviewer 1 Report

In this work, Dr. Mereu and colleagues investigate the role of EHMT2 in proteasome resistance, using two specific inhibitors (UNC0642 for most of the experiments, and then A366 in figure 5E only). This study has potential for clinical translation; however, the paper needs to be improved with additional experiments and better description of the experiments.

Major points:

- Is there any difference in protein and mRNA of EHMT2 in PIR-cell lines versus normal cell lines? What about EHMT1?

-  EHMT2 inhibitors:

o   How these inhibitors work? Why UNC0642 increases the protein levels of the EHMT2? What is the effect on EHMT1?

o   What is the IC50c of the single inhibitors? Why A366 is only used in one experiment?

- Genomic modulation of EHMT2 by si/shRNA should be performed to confirm the data obtained with the inhibitors

Minor points:

-      Doses of drugs used in the experiments should be reported in the main text or in the figure legends (e.g. what are sublethal doses of CFZ, for figure 3a?)

-      No SD, P-values, or number of replicates are reported (e.g. Figure 5A)

Author Response

We appreciate the Reviewer's insightful comments and suggestions, which have helped us to improve the quality of our manuscript. We have carefully addressed all the major and minor points and revised the manuscript accordingly.

Major points

  1. Is there any difference in protein and mRNA of EHMT2 in PIR-cell lines versus normal cell lines? What about EHMT1?

We thank the Reviewer for bringing up the possibility of EHMT1/EHMT2 upregulation in PI-resistant cells. To address this question, we compared EHMT1 and EHMT2 expression by RT-qPCR in PI-resistant and sensitive MM cell lines. We also explored EHMT2 expression at the protein level. As shown in Figures S2A and S2B and reported in lines 295-297 of the main text, our analyses did not detect any significant difference in expression between PI-resistant and sensitive MM cell lines. However, we regret to inform the Reviewer that we were unable to recruit a specific anti-EHMT1 antibody within the given timeframe to revise the manuscript. We apologize for any inconvenience and will properly explore this in future studies. 

  1. EHMT2 inhibitors:

2.1. How these inhibitors work?

We thank the reviewer for requiring us to describe the known mechanism of action of EHMT2 inhibitors used in the present study. We added a brief description in lines 88-91 (UNC0642) and lines 381-383 (A-366) of the revised manuscript.

2.2. Why UNC0642 increases the protein levels of the EHMT2?

We thank the reviewer for raising this issue. We unexpectedly observed that UNC0642 treatment induced EHMT2 protein accumulation in all cell lines tested. However, we do not have a clear explanation for this event. We demonstrated that EHMT2 upregulation did not occur at the mRNA level. In the discussion, we speculated that drug binding to EHMT2 would also inhibit its auto-methylation activity and suggested that this could lead to a possible stabilization of the EHMT2 protein. However, more studies are required to prove this hypothesis.

2.3. What is the effect on EHMT1?

It is known that EHMT1 and EHMT2 share 80% homology in methyltransferase SET domains. It was demonstrated that EHMT1 and EHMT2 heterodimerize to form a complex displaying overlapping functions in histone methyltransferase activity (Watson, Z.L. et al., 2019). Since UNC0642 is a specific EHMT1/2 inhibitor (Vinson, D.A. et al., 2022) we can speculate that it has a similar effect on both EHMT1 and EHMT2. EHMT1 mRNA expression data after UNC0642 treatment were added in Figure S5. 

2.4. What is the IC50c of the single inhibitors?

IC50 for CFZ and UNC0642 were calculated in PI-resistant and sensitive cell lines using cell growth luminescence assay. IC50 of PI-resistant and sensitive cells are now reported in Figure S3 of the revised manuscript.

2.5. Why A366 is only used in one experiment?

We thank the reviewer for raising this issue. In the revised version of the manuscript, we included additional experiments performed in PIR cell lines (Figure S6). We observed that the A366 inhibitor significantly synergized with CFZ in AMO-1 PIR cells already 72h post-treatment while 96h treatment was required to reach effects in the RPMI-8226 PIR cells. Notably, to achieve a similar synergistic effect with CFZ, a higher dose of A366 was required compared to UNC0642, therefore we decided to focus mainly on UNC0642.

  1. Genomic modulation of EHMT2 by si/shRNA should be performed to confirm the data obtained with the inhibitors

We thank the reviewer for the suggestion. To confirm the data obtained with UNC0642 we decided to use the alternative EHMT2 inhibitor A-366. A-366 has low structural similarity to UNC0642, however, it shows an analogous binding mode to EHMT2 and equivalent inhibition of H3K9 dimethylation (Milite C. et al., 2019). We observed that A-366 displayed synergistic activity with CFZ in PI-sensitive and PI-resistant, thus phenocopying UNC0642 action and supporting its specificity in inhibiting EHMT2/EHMT1 complex. 

We apologize to the reviewer if we do not show data using either siRNA or shRNA targeting EHMT2 to further support our pharmacological data. However, we should acknowledge that it has been recently shown that EHMT2 silencing reduced H3K9 dimethylation and carfilzomib-induced cell death of MM cells (DeSmedt et al 2021). Our study further supports the concept that EHMT2 inhibition could be a feasible strategy to increase proteasome inhibitor sensitivity also in drug-resistant MM patients. 

Minor points

  1. Doses of drugs used in the experiments should be reported in the main text or in the figure legends (e.g. what are sublethal doses of CFZ, for figure 3a?

As requested, we revised the main text and figure legends. For Figure 3A we added more detailed information in Figure S4. 

  1. No SD, P-values, or number of replicates are reported (e.g. Figure 5A)

We have added the information regarding the number of replicates, standard deviation, and p-values in the figure legends and/or in the main text where necessary. We apologize for the oversight and appreciate the reviewer bringing this to our attention. We are confident these revisions will enhance the clarity and transparency of the study's methodology and results.

Reviewer 2 Report

* The authors investigated a critical topic that will significantly contribute to the treatment of resistant multiple myeloma. The overall language of the manuscript is good and needs a moderate revision of the style and language.

* The title should not contain any abbreviation. Please, write EHMT2 entirely in the title.

* Line 34: MM needs to be defined in its first mention.

* The authors should make a more clear graphic abstract. Please, use the full name not a definition, and write the name of every structure below it. 

* All citations should be identified using numbers in square brackets.

* CZF in line 79 must be defined in its first mention.

* The author must write a full introduction about carfilzomib and its contribution to the multiple myeloma treatment.

* Line 99-101: if this is a previous result, please, put a reference.

* The authors must mention all catalog numbers for all used kits and model numbers for all used devices.

* Line 204: all primers should be put in a table with their sources and specific condition if present.

* Line 220: all primary antibodies should be put in a table with their sources and specific condition if present.

* Line 505: why did the authors use Figure S1 and Figure S2? Please, use them in the discussion section.

Author Response

We thank the Reviewer for the comments and for acknowledging the potential contribution of our findings to the treatment of multiple myeloma. We have carefully considered its suggestions and made the required revisions to improve the quality and clarity of our manuscript.

The title should not contain any abbreviation. Please, write EHMT2 entirely in the title.

We have revised the title to include the full name EHMT2.

Line 34: MM needs to be defined in its first mention.

We have defined MM in its first mention in the manuscript in line 51.

The authors should make a more clear graphic abstract. Please, use the full name not a definition, and write the name of every structure below it. 

Thank you for the suggestion. We have modified the graphical abstract to be more precise.

All citations should be identified using numbers in square brackets.

We have corrected the citation style.

CZF in line 79 must be defined in its first mention.

We have added the CFZ definition in line 59.

The author must write a full introduction about carfilzomib and its contribution to the multiple myeloma treatment.

We have added a brief description of CFZ's contribution to MM treatment in the introduction chapter.

Line 99-101: if this is a previous result, please, put a reference.

We have clarified the sentence in lines 111-112 and removed the need for a reference.

The authors must mention all catalog numbers for all used kits and model numbers for all used devices.

We have added catalog numbers and model numbers, as requested.

Line 204: all primers should be put in a table with their sources and specific condition if present.

We have designed a new table to list all primers used.

Line 220: all primary antibodies should be put in a table with their sources and specific condition if present.

We have designed a new table to list all antibodies used.

Line 505: why did the authors use Figure S1 and Figure S2? Please, use them in the discussion section.

We have clarified the use of Figure S1 and Figure S2 (now updated to S7) in the manuscript and mentioned them in line 110 and lines 405-407, respectively.

Reviewer 3 Report

With pleasure, I read the paper titled: “EHMT2 inhibition enhances carfilzomib sensitivity and overcomes drug resistance in multiple myeloma cell lines” by Mereu and colleagues. Overall, the subject matter is of clinical interest to a wide array of readers. The topic is intellectually relevant to the journal Cancers. Collectively, the manuscript reads well and has proper flow of ideas and data are summarized adequately in pertinent figures. The main strength of the paper includes being among the first to examine agents synergizing with carfilzomib in patients with MM. I have a few questions below:

1.      The rationale of the concentrations used is not clear. Have you done an MTT or a PrestoBlue assay to determine the IC50 value of UNC0642 before doing your experiments?

2.      Using colony formation and cell viability assays, have you examined the in-vitro safety of UNC0642 on non-cancerous (normal) cells, such normal human kidney cells (HEK239) or normal fibroblast cells (HS68 and BJ)? 

3.      Have you examined the antitumor effects of CFZ/UNC0642 combination on proliferation, migration, and invasion?

4.      Have you tested the antitumor effects of CFZ plus another chemotype of UNC0642 on MM cell lines? this is critical to validate your results.

5.      Have you validated if genetic inhibition (knockdown with shRNA or knockout with CRISPR/Cas9) of EHMT2 plus CFZ pharmacologic treatment result in similar synergistic anticancer effects?

6.      Figure 5 is interesting. Can you please explain why the protein level of EHMT2 was increased about treatment with UNC0642? Also, can you please provide a brief rationale as to why the protein expression level of H3K9me2 was significantly reduced upon treatment with 2 uM compared with 4 uM un HS-5 cells? Out of curiosity, how is the expression level for histone lysine demethylase subfamily 4 (KDM4) upon treatment with UNC0642; are they increased?

7.      Figure 6 talks about autophagy. Inhibition of mTOR pathway is central to activate autophagy. The authors should provide blots for some downstream targets of mTOR pathway to prove mTOR inhibition too place resulting in activation of autophagy. Examples of such proteins that can be probed include phospho-S6 (Ser235/236) and phospho-mTOR (Ser2448). You way want to provide the protein blot for total H2AX. Honestly, the antitumor effects of combination on CMYC are not dramatic. Please provide the concentrations used in this figure.

General questions (optional) that will substantially enhance the quality of your research.

1.      Have you done ‘unbiased’ pathway analysis (RNA-seq, for example) to examine the pathways enriched or depleted upon CFZ/UNC0642 combination treatment?

2.      If you knockdown or knockout EHMT2 in sensitive and/or resistant MM cell lines, will the cells become more/less sensitive to CFZ?

3.      Have you examined potential synergy between PNU-74654 and other standard-of-care chemotherapeutic agents?

4.      The study will become more significant if in-vivo xenograft data are included using monotherapy and combination therapy, as well as harvesting the xenografts at the end of experiment to examine WBC count and protein/mRNA profile. 

Author Response

We appreciate the Reviewer's insightful suggestions and for acknowledging the potential contribution of our findings to the treatment of multiple myeloma. We have carefully addressed all the points and made the required revisions to improve the quality of our manuscript.

1.The rationale of the concentrations used is not clear. Have you done an MTT or a PrestoBlue assay to determine the IC50 value of UNC0642 before doing your experiments? 

We thank the reviewer for requiring us to clarify the rationale of drug concentrations used in the present study. IC50 for CFZ and UNC0642 were calculated in PI-resistant and sensitive cell lines using cell growth luminescence assay. IC50 of PI-resistant and sensitive cells are now reported in Figure S3 of the revised manuscript. CFZ and UNC0642 sublethal concentrations were used for combinatorial treatments.

2.Using colony formation and cell viability assays, have you examined the in-vitro safety of UNC0642 on non-cancerous (normal) cells, such normal human kidney cells (HEK239) or normal fibroblast cells (HS68 and BJ)?

We thank the reviewer for raising the concern of UNC0642 in-vitro safety in normal cells. To address this issue we analyzed UNC0642 effects alone or in combination with CFZ on cell viability of normal cells such as peripheral blood mononuclear cells (PBMCs) from healthy donors and bone marrow-derived stromal cell line (HS5). FACS analysis of Annexin V-PI staining demonstrated a favorable cytotoxicity profile of single and combinatorial treatment in PBMCs (Fig 4A, Pink line) and in the bone marrow-derived stromal cell line HS-5 (Fig 4B, Pink line). To exclude that the bone marrow milieu could impair combinatorial treatment efficacy with the secretion of pro-survival factors, we co-cultured KMS28-BM myeloma cells on a layer of GFP-positive HS-5 cells. We demonstrated that UNC0642/CFZ combination progressively decreased the percentage of KMS28-BM cells, thus exhibiting selective cytotoxicity also in the presence of stromal cells (Fig. 4C). Moreover, using a 3D culturing system that efficiently reproduces multiple myeloma-bone marrow interactions we demonstrated that UNC0642/CFZ combination selectively killed MM cells (Fig 4D). Overall, these data support the safety of the suggested combinatorial treatment and encourage their use of in vivo preclinical models.

3.Have you examined the antitumor effects of CFZ/UNC0642 combination on proliferation, migration, and invasion? 

We thank the reviewer for raising this issue. Our research demonstrated that CFZ/UNC0642 combination strongly induces apoptosis in PI-resistant and PI-sensitive MM cells, as described in Figure 3A, S4, and 4D). Moreover, our results suggest that the cytotoxic effects of UNC0642/CFZ combination are not a consequence of cell cycle perturbation. We apologize to the reviewer if we did not explore the effects of CFZ/UNC0642 combination on migration and invasion. Further studies will be performed to properly address this relevant issue. However, we reasoned that any effect on cell migration and invasion could be a consequence of increased cell death.

4.Have you tested the antitumor effects of CFZ plus another chemotype of UNC0642 on MM cell lines? this is critical to validate your results. 

We thank the reviewer for the suggestion. To confirm the data obtained with UNC0642 we decided to use the alternative EHMT2 inhibitor A-366. A-366 has low structural similarity to UNC0642, however, it shows an analogous binding mode to EHMT2 and equivalent inhibition of H3K9 dimethylation (Milite C. et al., 2019). We observed that A-366 displayed synergistic activity with CFZ in PI-sensitive and PI-resistant MM cells (Figure 5E and S6), thus phenocopying UNC0642 action and supporting its specificity in inhibiting EHMT2/EHMT1 complex.

5.Have you validated if genetic inhibition (knockdown with shRNA or knockout with CRISPR/Cas9) of EHMT2 plus CFZ pharmacologic treatment result in similar synergistic anticancer effects?

We apologize to the reviewer if we do not show data using either siRNA or shRNA targeting EHMT2 to further support our pharmacological data. However, we should acknowledge that it has been recently described that EHMT2 silencing reduced H3K9 dimethylation and carfilzomib-induced cell death of MM cells (DeSmedt et al 2021). As previously mentioned, to confirm data obtained with UNC0642 we exploited the alternative EHMT2 inhibitor A-366 and demonstrated that it displayed synergistic activity with CFZ in PI-sensitive and PI-resistant MM cells (Figure 5E and S6). Overall, our study further supports the concept that EHMT2 inhibition could be a feasible strategy to increase proteasome inhibitor sensitivity also in drug-resistant MM patients.

6.Figure 5 is interesting. Can you please explain why the protein level of EHMT2 was increased about treatment with UNC0642? Also, can you please provide a brief rationale as to why the protein expression level of H3K9me2 was significantly reduced upon treatment with 2 uM compared with 4 uM un HS-5 cells? Out of curiosity, how is the expression level for histone lysine demethylase subfamily 4 (KDM4) upon treatment with UNC0642; are they increased? 

We thank the reviewer for raising these issues. We unexpectedly observed that UNC0642 treatment induced EHMT2 protein accumulation in all cell lines tested. However, we do not have a clear explanation for this event. We demonstrated that EHMT2 upregulation did not occur at the mRNA level. In the discussion, we speculated that drug binding to EHMT2 would also inhibit its auto-methylation activity and suggested that this could lead to a possible stabilization of the EHMT2 protein. However, more studies are required to prove this hypothesis. Considering H3K9me2 levels in HS-5 cells, we observed a discrete variability among experiments. However, we could appreciate that both concentrations of UNC0642 induced a significant H3K9me/H3K9me2 reduction compared to DMSO control cells. We regret that we do not have any data about KDM4 expression.

7.Figure 6 talks about autophagy. Inhibition of mTOR pathway is central to activate autophagy. The authors should provide blots for some downstream targets of mTOR pathway to prove mTOR inhibition too place resulting in activation of autophagy. Examples of such proteins that can be probed include phospho-S6 (Ser235/236) and phospho-mTOR (Ser2448). You way want to provide the protein blot for total H2AX. Honestly, the antitumor effects of combination on CMYC are not dramatic. Please provide the concentrations used in this figure.

We thank the reviewer for the suggestion. We analyzed the mTOR pathway by immunoblotting and added this information in Figure 6B. In AMO-1PIR cells, we noticed that inhibition of the mTOR pathway was mainly correlated with CFZ treatment, while LC3 accumulation was a consequence of EHMT2 inhibition. This may suggest that the two drugs work in parallel to synergistically increase the autophagic process. As requested, we provided a total H2AX protein blot in figure 6C. We would like to specify that c-MYC levels after UNC-0642 treatment in Figure 6A are different from Figure 5D because of the lower drug concentration used. We included drug concentrations for the experiments in the caption of Figure 6.

General questions (optional) that will substantially enhance the quality of your research.

1.Have you done ‘unbiased’ pathway analysis (RNA-seq, for example) to examine the pathways enriched or depleted upon CFZ/UNC0642 combination treatment? 

We thank the Reviewer for the suggestion. RNA-seq analysis will be explored in a follow-up study.

2.If you knockdown or knockout EHMT2 in sensitive and/or resistant MM cell lines, will the cells become more/less sensitive to CFZ?

We thank the Reviewer for the suggestion. We will properly assess this issue in a follow-up study.

3.Have you examined potential synergy between PNU-74654 and other standard-of-care chemotherapeutic agents?

Our primary screening performed in the U266PIR cell line included three compounds targeting the Wnt/β-catenin pathway [Foscenvivint (ICG-001), KYA1797K, and GNF-6231]. Among them, KYA1797K and ICG-001 showed high synergistic scores at two different concentrations in combination with CFZ. However, they either did not rank among the top synergistic compounds (KYA1797K), thus were not chosen for further validation, or they were not confirmed in the secondary screening carried out in additional 4 MM cell lines (ICG-001). A previous publication already identified a positive interaction between the β-Catenin inhibitor BC2059 and Bortezomib in Multiple Myeloma (Savvidou I et al 2017). Unfortunately, PNU-74654 was not included in the library of 320 small-molecule inhibitors. Thus, we regret we did not establish potential synergies between PNU-74654 and PI inhibitors. 

4.The study will become more significant if in-vivo xenograft data are included using monotherapy and combination therapy, as well as harvesting the xenografts at the end of experiment to examine WBC count and protein/mRNA profile.

We thank the Reviewer for the suggestion. We will properly assess this issue in a follow-up study.

Round 2

Reviewer 1 Report

No further comments